# Common genetic variations in telomere length genes and lung cancer: a Mendelian randomisation study and its novel application in lung tumour transcriptome

Ricardo Cortez Cardoso Penha[1]*, Karl Smith-Byrne[2], Joshua R Atkins[1], Philip C Haycock[3], Siddhartha Kar[3], Veryan Codd[4,5], Nilesh J Samani[4,5], Christopher Nelson[4,5], Maja Milojevic[1], Aurélie AG Gabriel[6], Christopher Amos[7], Paul Brennan[1], Rayjean J Hung[8], Linda Kachuri[9], James D Mckay[1]*

[1]Genomic Epidemiology branch, International Agency for Research on Cancer/World Health Organization (IARC/WHO), Lyon, France; [2]Cancer Epidemiology Unit, University of Oxford, Oxford, United Kingdom; [3]MRC Integrative Epidemiology Unit, Bristol Population Health Science Institute, Bristol Medical School (PHS), Bristol, United Kingdom; [4]Department of Cardiovascular Sciences, University of Leicester, Leicester, United Kingdom; [5]NIHR Leicester Biomedical Research Centre, Glenfield Hospital, Leicester, United Kingdom; [6]Ludwig Lausanne Branch, Faculty of Biology and Medicine, Lausanne, Switzerland; [7]Institute for Clinical and Translational Research, Baylor College of Medicine, Houston, United States; [8]Lunenfeld-Tanenbaum Research Institute, Sinai Health, Toronto, Canada; [9]Departament of Epidemiology and Population Health, Stanford University, Stanford, United States

*For correspondence:
cortezr@iarc.who.int (RCCP);
mckayj@iarc.who.int (JDM)

**Competing interest:** The authors declare that no competing interests exist.

## Abstract

**Background:** Genome-wide association studies (GWASs) have identified genetic susceptibility variants for both leukocyte telomere length (LTL) and lung cancer susceptibility. Our study aims to explore the shared genetic basis between these traits and investigate their impact on somatic environment of lung tumours.

**Methods:** We performed genetic correlation, Mendelian randomisation (MR), and colocalisation analyses using the largest available GWASs summary statistics of LTL (N=464,716) and lung cancer (N=29,239 cases and 56,450 controls). Principal components analysis based on RNA-sequencing data was used to summarise gene expression profile in lung adenocarcinoma cases from TCGA (N=343).

**Results:** Although there was no genome-wide genetic correlation between LTL and lung cancer risk, longer LTL conferred an increased risk of lung cancer regardless of smoking status in the MR analyses, particularly for lung adenocarcinoma. Of the 144 LTL genetic instruments, 12 colocalised with lung adenocarcinoma risk and revealed novel susceptibility loci, including *MPHOSPH6*, *PRPF6*, and *POLI*. The polygenic risk score for LTL was associated with a specific gene expression profile (PC2) in lung adenocarcinoma tumours. The aspect of PC2 associated with longer LTL was also associated with being female, never smokers, and earlier tumour stages. PC2 was strongly associated with cell proliferation score and genomic features related to genome stability, including copy number changes and telomerase activity.

**Conclusions:** This study identified an association between longer genetically predicted LTL and lung cancer and sheds light on the potential molecular mechanisms related to LTL in lung adenocarcinomas.

**Funding:** Institut National du Cancer (GeniLuc2017-1-TABAC-03-CIRC-1-TABAC17-022), INTE-GRAL/NIH (5U19CA203654-03), CRUK (C18281/A29019), and Agence Nationale pour la Recherche (ANR-10-INBS-09).

## Editor's evaluation

This study is of interest to epidemiologists and geneticists studying the association between telomere length and lung cancer risk. This work provides useful insight into risk factors for lung cancer. Overall, the results of this study are solid, as the genetic instrument used here is better powered and the battery of Mendelian randomization analysis makes this broad set of results convincing compared to previous work.

## Introduction

Telomeres are a complex of repetitive TTAGGG sequences and nucleoproteins located at the end of chromosomes and have an essential role in sustaining cell proliferation and preserving genome integrity (*de Lange, 2009*). Telomere length progressively shortens with age in proliferative somatic cells due to incomplete telomeric regions replication (*Watson, 1972*) and low activity of the telomerase *TERT* in adult cells. The shortening of the telomere length results in cell cycle arrest, cellular senescence, and apoptosis in somatic cells (*Harley et al., 1990*). The maintenance of telomere length, which allows cancer cells to escape the telomere-mediated cell death pathways, is one feature related to the hallmarks of cancer (*Hanahan and Weinberg, 2011*).

Telomere length appears to vary between individuals and has been studied in relation to many diseases. In observational studies, telomere length is measured as the average length of telomeric sequences in a given tissue (*Montpetit et al., 2014*). Telomere length appears correlated across tissue types (*Demanelis et al., 2020*), and as such, leukocyte telomere length (LTL) is generally measured in epidemiologic studies as a proxy for telomere length in other tissues. Recently, LTL has been measured in 472,174 individuals from the UK Biobank (UKBB; *Codd et al., 2021*), and LTL was associated with multiple biomedical traits (i.e. pulmonary and cardiovascular diseases, haematological traits, lymphomas, kidney cancer, and other cancer types). Genetic analysis of LTL also revealed 138 genetic loci linked to LTL across a variety of different genes involved in telomere biology and DNA repair (*Codd et al., 2021*).

In the context of lung cancer, genetic variants at several loci have been associated with both LTL and lung cancer risk, including variants near the *TERT*, *TERC*, *OBFC1*, and *RTEL1* genes, fundamental to telomere length maintenance (*McKay et al., 2008*; *Wang et al., 2008*; *Rafnar et al., 2009*; *Kachuri et al., 2016*; *McKay et al., 2017*). The effects of the telomere-related variants appear more relevant to lung adenocarcinoma risk than other histologic subtypes (*McKay et al., 2017*; *Landi et al., 2009*). Accordingly, a causal relationship between LTL and susceptibility to lung cancer was observed using Mendelian randomisation (MR) approaches (*Zhang et al., 2015*; *Haycock et al., 2017*; *Kachuri et al., 2019*) as well as in observational studies that have associated directly measured telomere length with risk of lung cancer (*Sanchez-Espiridion et al., 2014*; *Zhang et al., 2017*).

The aim of the current work was to investigate the relationship between genetically predicted LTL and lung cancer, including lung cancer histological subtypes and smoking status. To this end, we conducted genome-wide correlations, MR, and colocalisation analyses to explore the relationship between LTL and lung cancer. We additionally undertook polygenic risk score (PRS) analysis using the LTL genetic instrument to explore the influence of LTL on the demographic, clinical, and molecular features of lung adenocarcinoma tumours.

# Materials and methods

## Reporting guidelines

The current study has been reported according to the STROBE-MR guidelines (Reporting Standards Document).

## Data

Genome-wide association studies (GWASs) summary statistics for lung cancer (29,239 cases and 56,450 controls) and stratified by histological subtype (squamous cell carcinoma, small-cell carcinoma, and adenocarcinoma) and smoking status (ever and never smokers) were obtained from the International Lung Cancer Consortium (ILCCO; *McKay et al., 2017*). All analyses of LTL requiring summary statistics used results from a GWAS of LTL in 464,716 individuals of European ancestry from the UKBB (*Codd et al., 2021*). Downstream analyses considered additional lung cancer risk factors, such as lung function and cigarette smoking. We obtained GWAS summary statistics for forced expiratory volume in 1 s (FEV$_1$) and forced vital capacity (FVC) from a published UKBB analysis (*Kachuri et al., 2020*). For smoking behaviour traits, we used results from the GWAS and Sequencing Consortium of Alcohol and Nicotine use (GSCAN) consortium meta-analysis of cigarettes per day (continuous), smoking initiation (ever versus never), smoking cessation (successfully quit versus continuing), and age at smoking initiation (continuous; *Liu et al., 2019*) excluding the UKBB participants. For the obesity-related traits (continuous), we used the results from the UKBB and GIANT meta-analysis of BMI (*Pulit et al., 2019*) and waist-to-hip ratio (WHR; *Pulit et al., 2019*), or OpenGWAS data using UKBB participants (*Elsworth et al., 2020*) for high-density lipoprotein (HDL), triglycerides, and systolic and diastolic blood pressure. For the alcohol behaviour trait, we obtained the results from GSCAN phase 2 of drinks per week (continuous; *Saunders et al., 2022*). Colocalisation analyses of gene expression used lung tissue expression quantitative trait loci (eQTL) summary statistics from the Genotype-Tissue Expression (GTEx) data version 8.

Analyses of molecular phenotypes were performed using 343 lung adenocarcinoma samples of European ancestry from The Cancer Genome Atlas (TCGA) cohort with germline profile, RNA-sequencing, and epidemiological data available. Genotyping and imputation of germline variants have been described elsewhere (*Gabriel et al., 2022*). The total somatic mutation burden of TCGA samples was obtained from *Ellrott et al., 2018*, and DNA mutational signatures were extracted and attributed, as previously described (*Gabriel et al., 2022*). RNA-sequencing data were obtained from TCGA data portal using TCGA biolinks package in R (version 2.22.3; *Colaprico et al., 2016*). Telomere length measurement by whole-genome sequencing (WGS-measured TL, 655 samples across cancer sites) was retrieved from *Barthel et al., 2017*.

Tumour genomic characteristics were defined by the analyses of the TCGA data, including gene expression-based scores of telomerase activity (*Barthel et al., 2017*) and cellular proliferation (*Thorsson et al., 2018*), as well as the observed frequency of somatic homologous recombination-related events (represented as a homologous recombination repair deficiency score), and the average number of somatic copy number alteration within the tumours (*Knijnenburg et al., 2018*).

## Linkage disequilibrium score regression

Genetic correlations across traits were calculated using linkage disequilibrium score regression (LDSC) by the LDSC package (v1.0.0; *Bulik-Sullivan et al., 2015*). Linkage disequilibrium (LD) scores were generated on the 1000 Genomes Project Phase 3 reference panel with the HLA region excluded as provided by the package due to long range LD patterns. The genome-wide correlations that passed Bonferroni correction (adjusted p-values<0.05) were considered statically significant.

## Mendelian randomisation

MR is a method for interrogating relationships between putative risk factors and health outcomes by using genetic variants associated with the exposure of interest, typically obtained from GWAS, as instrumental variables. Assuming that fundamental MR assumptions are satisfied, this approach can be said to identify unbiased causal estimates. The genetic instrument for LTL was defined as the set of 144 genetic variants that were genome-wide significant (p<5x10$^{-08}$) but not in linkage disequilibrium with each other (r$^2$<0.01) and restricted to common genetic variation (minor allele frequency >1%) in European populations. Proxy variants in LD (r$^2$>0.8, whenever possible) were chosen when a genetic

variant was not available in the lung cancer GWAS. Primary MR analyses were conducted using the inverse-variance method with multiplicative random-effects (*Yavorska and Burgess, 2017*). Sensitivity analyses to horizontal pleiotropy and other violations of MR assumptions were performed using other MR estimation methods, such as weighted median, MR-Egger, contamination mixture model, MR-PRESSO, and MR-RAPS (*Yavorska and Burgess, 2017*; *Sanderson et al., 2022*). Multivariable MR (MVMR) methods included the inverse-variance weighted, MR-Egger, and least absolute shrinkage and selection operator (LASSO)-based methods (*Yavorska and Burgess, 2017*; *Sanderson et al., 2019*).

## Colocalisation methods

Unlike MR, where the goal is to assess the evidence for a causal effect of an exposure on an outcome, colocalisation is agnostic with respect to direction of effect and only assesses the probability that the two traits are affected by the same genetic variants at a given locus. Colocalisation can be viewed as a complementary approach for evaluating MR assumptions within specific genes or regions since strong evidence of colocalisation indicates overlap in genetic mechanisms affecting LTL and lung cancer. We used COLOC (v5.1.0; *Wallace, 2020*) to estimate the posterior probability for two traits sharing the same causal variant ($PP_4$) in a 150 kb LD window, with $PP_4 > 0.70$ corresponding to strong evidence of colocalisation, as previously suggested (*Wallace, 2020*; *Lopes et al., 2022*). Priors chosen for the colocalisation analyses were $p1=10^{-3}$, $p2=10^{-4}$, and $p12=10^{-5}$, or approximately, a 75% prior belief that a signal will only be observed in the LTL GWAS and less than 0.01% prior belief in favour of colocalisation between the two traits at a given locus (*Giambartolomei et al., 2014*). Conditioning and masking colocalisation methods were also used as they may identify putative shared causal variants in the presence of multiple causal variants present in a defined LD window (*Wallace, 2021*). We present the average $PP_4$ from all methods as our posterior belief in favour of colocalisation between LTL and lung cancer risk. Multi-trait colocalisation based on a clustering algorithm was also performed using HyPr-Coloc (v1.0) to identify shared genetic signals with other lung cancer-related traits (*Foley et al., 2021*).

## Principal component analysis based on RNA-sequencing data

Read counts of RNA-sequencing data were normalised within (GC-content and gene length) and between (sequencing depth) lane procedures by EDASeq R package (version 2.28.0; *Risso et al., 2011*) and excluding low read counts. Principal component analysis was applied using singular value decomposition method, after excluding extreme outliers. Pathway analyses were conducted using Gene Set Enrichment Analysis software (GSEA, version 4.2.3; *Subramanian et al., 2005*) on gene annotations from Gene Ontology database. Pathway analyses were restricted to the top 500 genes positively and negatively correlated with each principal component that passed multiple-testing correction (Bonferroni-adjusted p-value<0.05 for 74,465 tests), which is the maximum number of genes supported by the online software.

The PRS for LTL was composed of the same 144 variants used in the MR analysis and was computed as the sum of the individual's beta-weighted genotypes using PRSice-2 software (*Choi and O'Reilly, 2019*). Associations were estimated per SD increase in the PRS, which was normalised to have a mean of zero across lung adenocarcinoma samples of European ancestry within the TCGA cohort. The associations between the eigenvalues of the gene expression principal components (outcome) and demographic, clinical, and genomic features related to genome stability (predictors derived from TCGA published papers and TCGA data portal, except for the DNA mutational signatures [*Gabriel et al., 2022*]) were calculated using a multivariate linear regression model.

## Inferring PC2 gene expression signature based on RNA-sequencing data

The TCGA-lung adenocarcinoma (LUAD) tumour samples were split into training (70%, N=255) and validation (30%, N=108) datasets. Principal components analysis based on RNA-sequencing data was performed to summarise the gene expression profiles of lung adenocarcinoma tumours into five principal components in the training and validation datasets separately, as previously described (see 'Principal component analysis based on RNA-sequencing data' section in methods). Subsequently, we applied the partial least squares-based method called rigid transformation (*Hubert and Branden, 2003*) to align the first five principal components in both datasets. This method compares embeddings,

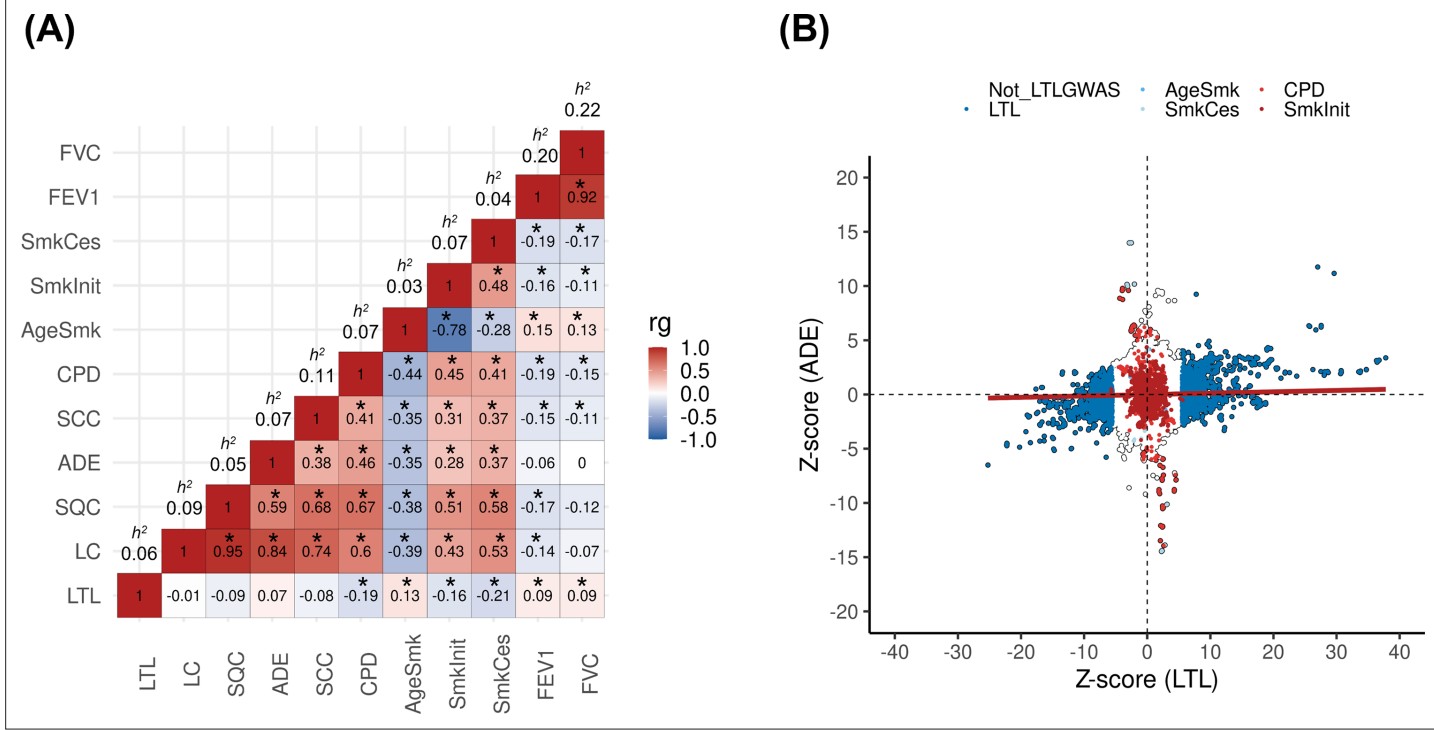

**Figure 1.** Genetic correlations between leukocyte telomere length (LTL) and lung cancer (LC) related traits. (**A**) Heatmap representing the genetic correlation analyses (rg) for LTL across LC, histological subtypes (lung adenocarcinoma [ADE], squamous cell carcinoma [SQC], and small-cell carcinoma [SCC]), smoking propensity (cigarettes per day [CPD], smoking cessation [SmkCes], Smoking initiation [SmkInit], and age of smoking initiation [AgeSmk]), and lung function related (forced vital capacity [FVC] and forced expiratory volume [FEV1]) traits. The black star indicates correlations that passed Bonferroni correction ($p < 4 \times 10^{-04}$). Heritability ($h^2$) as the proportion of the phenotypic variance caused by SNPs. (**B**) Plot of Z-scores (ADE versus LTL), restricting to the Hapmap SNPs (~1.2 million) but excluding HLA region. Genome-wide significant SNPs ($p < 5 \times 10^{-08}$) for each trait were coloured (CPD in red, SmkInit in dark red, LTL in dark blue, AgeSmk in blue, SmkCes in lightblue, and not genome-wide hits for LTL or any other selected trait in white). Linear regression line was coloured in red.

The online version of this article includes the following figure supplement(s) for figure 1:

**Figure supplement 1.** Design of the study.

low-dimensional representations in both datasets, performing a slightly rotation of principal components in order to translate and match them in both training and validation datasets. To select the most informative genes of PC2 in the training dataset, RNA levels of the genes correlated with PC2 (N=3914 out of 14,893 genes, FDR <0.05 for 14,893 tests) were selected as variables for the LASSO regression models. The LASSO tune parameters were chosen by resampling the training dataset (1000 bootstraps: root mean of SE=0.12 ± 0.0004, Lambda = 10 x 10$^{-10}$, r$^2$=0.99 ± 0.00007) using the tidymodels metapackage in R (v1.0.0; wrapper of glmnet). The 10 genes selected by the LASSO model were used to infer the gene expression signature of PC2 by adding up the scaled values of the log-normalised read counts of each gene multiplied by the respective LASSO regression coefficients. For validation purpose, the inferred PC2 signature was calculated in the validation dataset and compared with the observed principal components. After validation in the subset of the TCGA-LUAD cohort, the inferred PC2 signature was calculated in TCGA-LUSC dataset to compare differences between lung cancer histological subtypes.

## Results

### Genome-wide genetic correlations

The design of the study is represented in *Figure 1—figure supplement 1*. We first assessed the shared genetic basis of telomere length, lung cancer risk, and other putative lung cancer risk factors, such as smoking behaviours (age start smoking, smoking cessation, smoking initiation, and cigarettes per day) and lung function (FEV$_1$ and FVC) using genome-wide correlations (*Figure 1A*). There was little

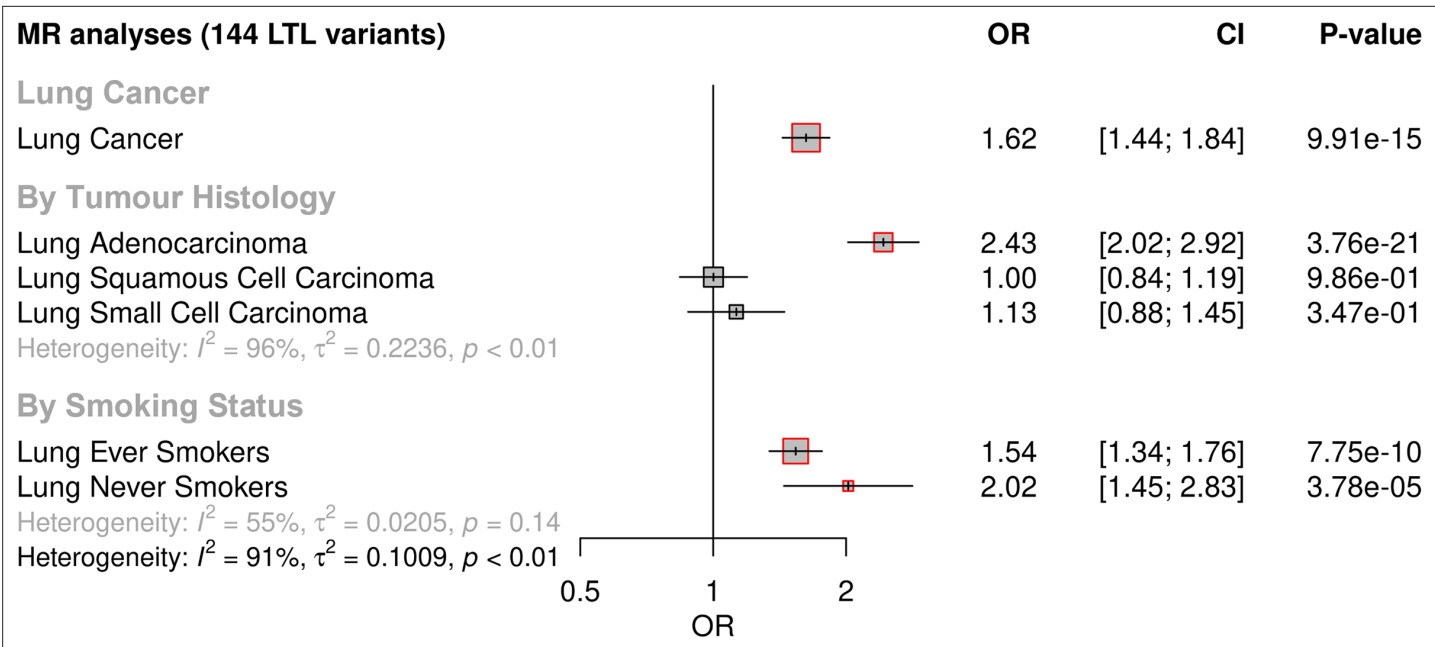

**Figure 2.** Genetically predicted leukocyte telomere length (LTL) association with lung cancer. Lung cancer (by histology or by smoking status) risk associations with the LTL instrument from the inverse-variance-weighted MR analyses are expressed as OR per SD increase in genetically predicted LTL. Statistically significant associations with p-values<0.05 (red square). Heterogeneity is estimated by the statistic $I^2$, tau variance of subgroups ($\tau^2$), and p-values for Cochran's Q heterogeneity measure.

The online version of this article includes the following figure supplement(s) for figure 2:

**Figure supplement 1.** Sensitivity analysis of the genetically predicted leukocyte telomere length (LTL) Mendelian randomisation (MR) instrument.

evidence for genetic correlations by LDSC between LTL variants and lung cancer ($r_g$ = −0.01, p=0.88) or when stratified by histologic subtypes (*Figure 1A*). Increasing LTL was genetically correlated with older age at smoking initiation ($r_g$ = 0.13, p=3.0 × 10$^{-3}$), and negatively correlated with smoking cessation: ($r_g$ = −0.21, p=6.9 × 10$^{-09}$), smoking initiation ($r_g$ = −0.16, p=1.3 × 10$^{-10}$), and cigarettes per day ($r_g$ = −0.19, p=2.1 × 10$^{-08}$). Longer LTL was genetically correlated with improved lung function, as indicated by increasing values of FEV$_1$ ($r_g$ = 0.09, p=5.1 × 10$^{-07}$) and FVC ($r_g$ = 0.09, p=1.1 × 10$^{-05}$). To better understand the absence of genome-wide correlations between LTL and lung cancer, we visualised the Z-scores for each trait for approximately 1.2 million variants included in the LDSC analyses (*Figure 1B*). A subgroup of variants associated with longer LTL was correlated with increased lung adenocarcinoma risk, while the subgroup of smoking-behaviour associated variants, which also conferred an increased risk of lung adenocarcinoma, tended to have lower LTL.

## MR analyses

From the 490 genetic instruments associated with LTL at genome-wide significance (p<5x10$^{-08}$), 144 LTL genetic instruments, that explained ~3.5% of the variance in LTL, and were in low-linkage disequilibrium ($r^2$<0.01) were used in MR analysis (*Supplementary file 1a*). As a sensitivity analysis, a PRS composed of these genetic instruments was associated with TL estimated from WGS in blood samples across TCGA cohorts (Beta = 0.03, 95%CI = 0.01–0.05, p=0.001) but was not associated with TL in tumour material from the same patients (*Figure 2—figure supplement 1*).

MR analyses demonstrated that longer genetically predicted LTL was associated with increased lung cancer risk (OR = 1.62, 95% CI = 1.44–1.84, p=9.91 × 10$^{-15}$) (*Figure 2*; *Supplementary file 1*). Longer LTL conferred the largest increase in risk for lung adenocarcinoma tumours (OR = 2.43, 95% CI = 2.02–2.92, p=3.76 × 10$^{-21}$), but there was limited evidence of a causal relationship for other histologic subtypes, such as squamous cell carcinoma (OR = 1.00, 95% CI = 0.84–1.19, p=0.98) and small-cell carcinoma (OR = 1.13, 95% CI = 0.87–1.45, p=0.34; *Figure 2*, *Supplementary file 1*). However, our study was underpowered to detect an association between lung small-cell carcinoma and LTL at OR of 1.13 and considering alpha type-1 error rate of 5% (*Figure 2—figure supplement 1*). When

stratifying the analyses by smoking status, LTL was associated with lung cancer risk in both never (OR = 2.02, 95% CI = 1.45–2.83, p=3.78 × 10⁻⁰⁵) and ever smokers (OR = 1.54, 95% CI = 1.34–1.76, p=7.75 × 10⁻¹⁰; *Figure 2*, *Supplementary file 1*). Evidence for negative pleiotropy (*supplementary file 1c*) and heterogeneity (*supplementary file 1d*) was observed for all lung cancer outcomes except for squamous cell carcinoma. However, a significant association for LTL and lung cancer risk was found for methods robust to the significant directional pleiotropy (MR-Egger: lung cancer [OR = 2.35, p=3.37 × 10⁻¹³]; lung adenocarcinoma [OR = 4.48, p=7.30 × 10⁻¹⁷]; never smokers [OR = 6.84, p=2.07 × 10⁻¹⁰]; *supplementary file 1b*). Leave-one-out analyses detected only one outlier, rs7705526 in *TERT*, resulting in >10% change in MR effect size for associated lung cancer subtypes (*supplementary file 1e*). MVMR analyses considering instruments related to LTL and WHR, HDL, total triglycerides, systolic

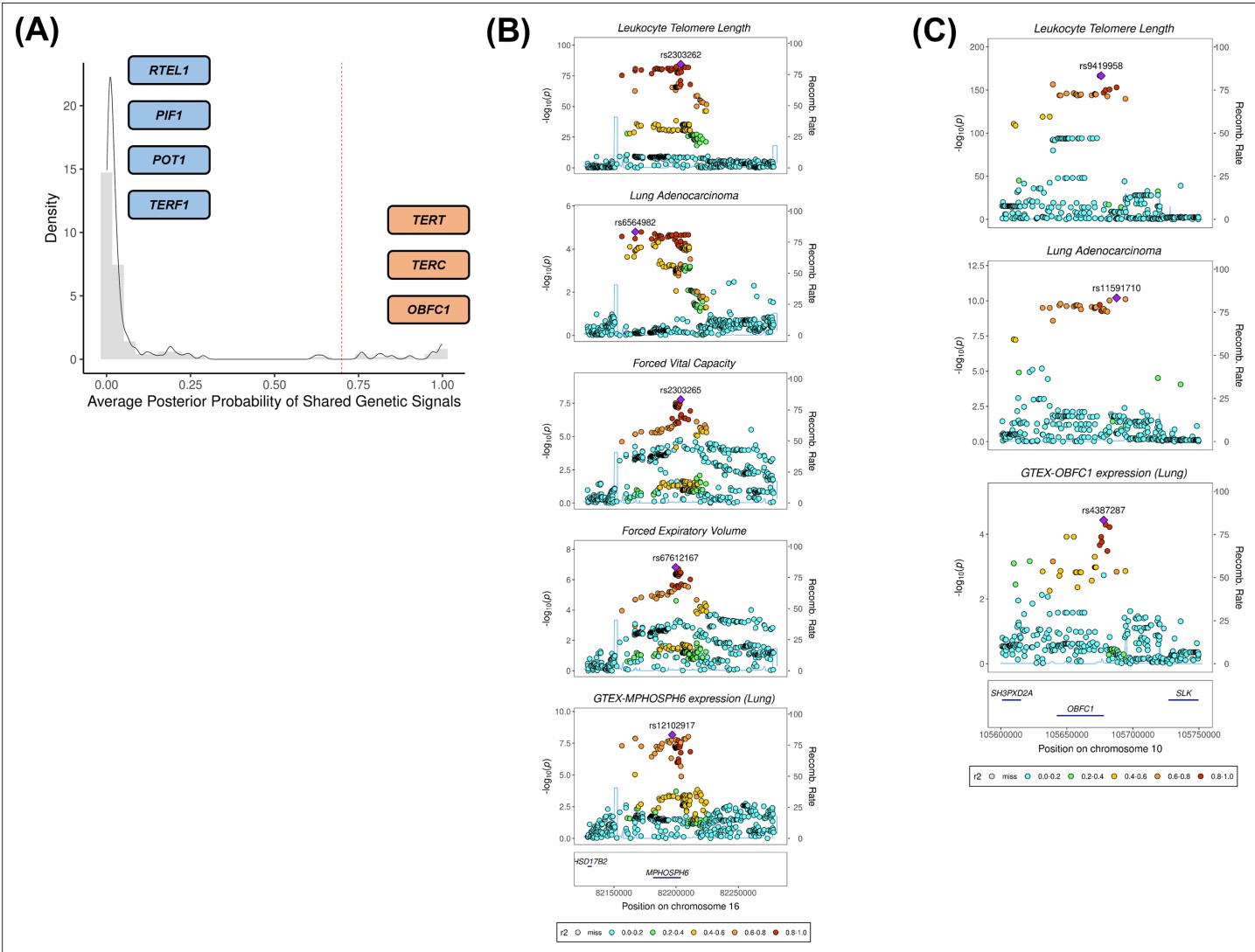

**Figure 3.** Colocalisation analyses for the genetic loci defined by the 144 leukocyte telomere length (LTL) variants. (**A**) Distribution of the average posterior probability for shared genetic loci between LTL and lung adenocarcinoma, highlighting in orange the telomere maintenance loci that colocalised (avg_PP4≥0.70) and in blue the ones where there was limited evidence for colocalisation (avg_PP4<0.70). Dashed red line represents the arbitrary avg_PP4 cutoff of 0.70. Representative stack plots for the multi-trait colocalisation results within (**B**) *MPHOSPH6* and (**C**) *OBFC1* loci, centred on a 150 kb LD window of rs2303262 and rs9419958 variants, respectively. Left Y-axis represents the –log10(p-values) of the association in the respective genome-wide association study for a given trait. The right Y-axis represents the recombination rate for the genetic loci. The X-axis represents the chromosome position. SNPs are coloured by the linkage disequilibrium correlation threshold (**r2**) with the query labelled SNP in European population. Sentinel SNPs within the defined LD window were labelled in each trait.

The online version of this article includes the following figure supplement(s) for figure 3:

**Figure supplement 1.** Association plots for leukocyte telomere length (LTL) and lung adenocarcinoma at *RTEL1* locus.

blood pressure, smoking, and alcohol intake, as well as multiple traits combined, suggested that the association between LTL and lung adenocarcinoma risk is independent of smoking propensity, obesity-related, and alcohol intake-related traits (*supplementary file 1f, 1g*).

## Colocalisation analyses

We investigated whether there was evidence of shared genetic signals between LTL and lung adenocarcinoma at loci centred on the 144 genetic instruments used in MR analyses using colocalisation methods (*Figure 3A*, *supplementary file 1h*). Loci with evidence of colocalisation between LTL and lung adenocarcinoma tended to be near genes that encode telomerase subunits and its associated complex, including genetic variants at *TERT* (5p15.33; rs116539972, rs7705526, rs61748181, rs71593392, and rs140648021), *TERC* (3q26.2; rs12638862 and rs146546514), and *OBFC1* (10q24.33; rs9419958 and rs139122544). Several colocalised loci mapped to genes that have not been previously linked to lung cancer risk at genome-wide significant level: *MPHOSPH6* (16q23.3; rs2303262), *PRPF6* (20q13.33; rs80150989), and *POLI* (18q21.2; rs2276182). Other telomere maintenance genes showed limited evidence of colocalisation with lung adenocarcinoma (i.e. *TERF1* and *PIF1*). For instance, while the *RTEL1* locus (20q13.33: rs117238689, rs115610405, rs35640778, and rs35902944) harboured variants associated with both LTL and lung adenocarcinoma (*Figure 3—figure supplement 1*), these signals appeared to be distinct and independent of each other (avg_PP3=0.999, avg_PP4=0.001; *Figure 3A*, *supplementary file 1h*).

We further evaluated whether the loci colocalised between LTL and lung adenocarcinoma also shared genetic signals with other traits related to lung cancer susceptibility (*supplementary file 1i*). Multi-trait analyses at the 16q23.3 locus colocalised rs2303262 with *MPHOSPH6* expression in lung tissue, FVC and FEV$_1$, but not with any of the traits related to smoking behaviour (p=0.72; *Figure 3B*, *supplementary file 1i*). We additionally identified evidence of colocalisation (p=0.74) between lung adenocarcinoma, LTL, and gene expression in lung epithelial cells for two variants at the *OBFC1* locus: rs139122544 and rs9419958 (*Figure 3C*, *supplementary file 1i*).

## Genetically predicted LTL association with tumour features

We investigated the impact of genetically predicted LTL on lung adenocarcinoma tumour features by estimating molecular expression patterns within 343 lung adenocarcinomas tumours using principal component analysis in RNA-sequencing data. The first five components explained ~54% of the observed variance in the RNA-sequencing data (*Figure 4*, *Figure 4—figure supplement 1*). To explore the biological meaning of the five components, we performed pathway analyses for the top 500 genes with the highest loadings in each component (*supplementary file 1j*, *supplementary file 1k*). Overall, the genes correlated with each component tended to be enriched for specific cell signaling pathways (PC1: RNA processing; PC2: cell-cycle; PC3: metabolic processes; PC4: immune response; PC5: cellular response to stress and DNA damage; false discovery rate <5%; *supplementary file 1l*).

We then tested the association between the PRS composed of the 144 genetic instruments selected for MR analysis and the five components of gene expression within lung adenocarcinoma tumours (*Figure 4A*). The LTL PRS was positively associated with the second component (PC2) of tumour expression (Beta = 0.17, 95% CI = 0.12–0.19, p=$1.0 \times 10^{-3}$; *Figure 4A*). In multivariate analysis, higher values of PC2 tended to be associated with patients older at diagnosis (p=0.001), female (p=0.005), being never smokers (p=0.04), and diagnosed with early-stage tumours (p=0.002; *Table 1*). PC2 was also highly correlated with gene expression-based measure of cell proliferation and several genomic features related to genomic stability (*Figure 4B*). In multivariate analysis, higher values of PC2 were associated with reduced tumour proliferation (p=$3.7 \times 10^{-30}$), lower somatic copy number alternations (p=$1.6 \times 10^{-05}$), and higher tumour telomerase activity scores (p=$1.6 \times 10^{-5}$). Multivariate analysis also indicated that LTL PRS remained an independent predictor of PC2 when considering these genomic features (p=0.009; *Table 1*). It is noteworthy only nominal associations between LTL PRS and above-mentioned features and none remained statistically significant after correction for multiple testing (*supplementary file 1m*). We next inferred the gene expression signature of PC2, based on 10 genes informative of this component selected by the LASSO regression models, in both lung adenocarcinoma (TCGA-LUAD) and squamous cell carcinoma (TCGA-LUSC) cohorts (*Figure 5*, *Figure 5—figure supplement 1*). The association between LTL PRS and inferred PC2 was observed in TCGA-LUAD (p=0.001) but not in TCGA-LUSC (p=0.729) cases (*Figure 5A*). The inferred PC2 signature levels were

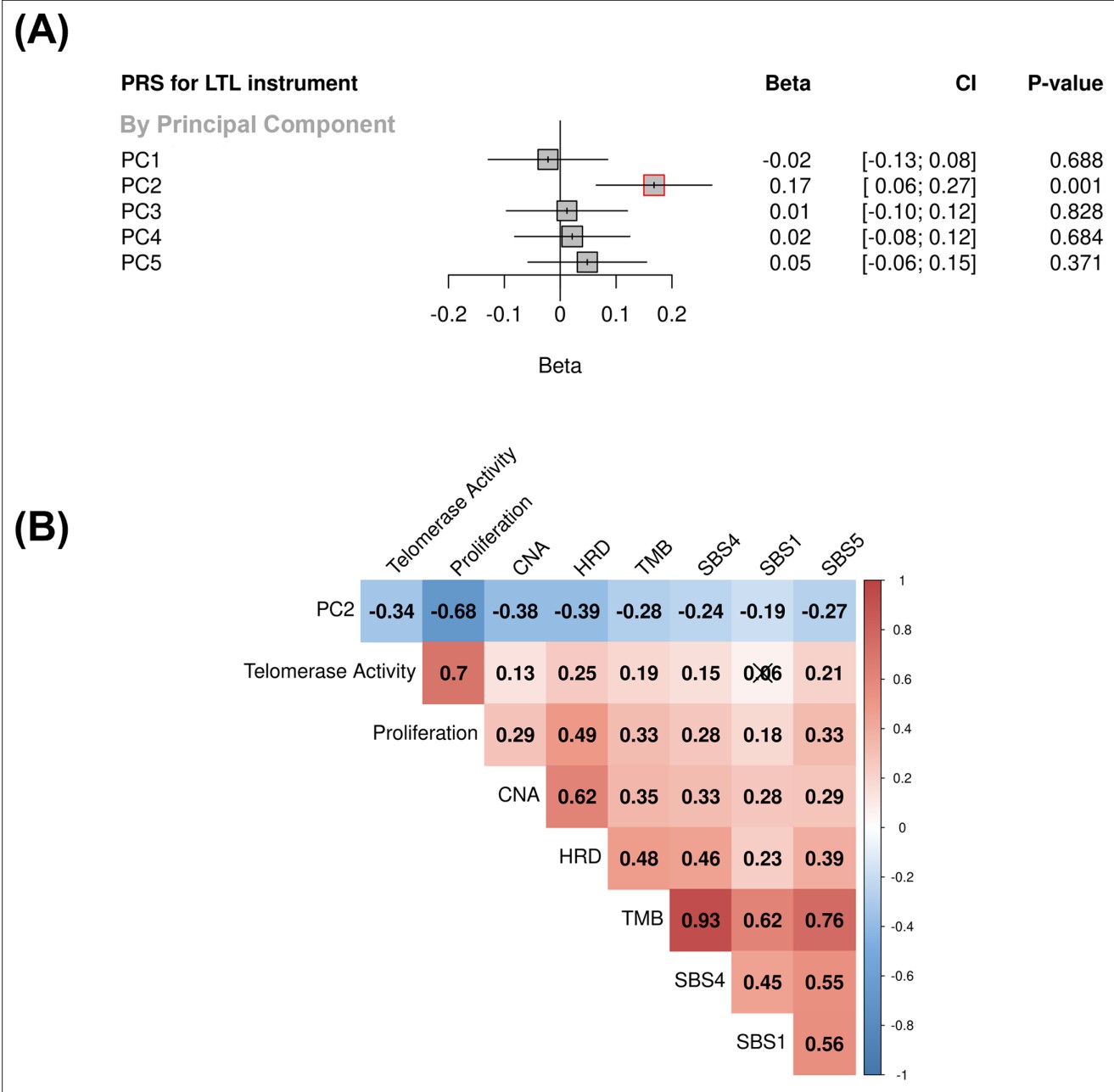

**Figure 4.** Associations between molecular expression patterns of lung adenocarcinoma tumours, LTL PRS, and The Cancer Genome Atlas (TCGA) features. (**A**) LTL PRS association with the first five principal components based on RNA-sequencing data of lung adenocarcinomas tumours (n=343). Results are expressed as beta estimate per SD increase in genetically predicted LTL. Linear regression model adjusted by sex, age, smoking status, and PC1-5 (genetic ancestry) covariates. Statistically significant associations with p-values<0.05 (red square). (**B**) Heatmap representing the correlations among PC2 and selected molecular features related to telomere length canonical roles. LTL = leukocyte telomere length; PRS = polygenic risk score; PC = principal component; TMB = tumour total mutation burden; HRD = homologous recombination deficiency, SBS (single base substitution DNA mutational signatures). SBS1 and SBS5 are DNA mutational signatures associated with age-related processes, and SBS4 is associated with tobacco smoking exposure. X-shaped marker to cross correlations with p-value>0.05.

The online version of this article includes the following figure supplement(s) for figure 4:

**Figure supplement 1.** Principal component analysis (PCA) based on RNA-sequencing data.

**Table 1.** Association between PC2 (outcome) and lung adenocarcinoma tumour features in univariate and multivariate models (n=343).

Non-molecular features

| Predictors | Univariate model | | Multivariate model | |
|---|---|---|---|---|
| | OR/Beta (SE) | p-value | OR/Beta (SE) | p-value |
| Age at diagnosis[*] | 0.17±0.05 | 0.001 | 0.17±0.05 | 0.001 |
| Gender (male)[†] | 0.73±0.11 | 0.005 | 0.74±0.11 | 0.005 |
| Smoking status (ever)[†] | 0.67±0.16 | 0.013 | 0.72±0.15 | 0.035 |
| Tumour stage (late)[†] | 0.67±0.13 | 0.002 | 0.67±0.13 | 0.002 |

Molecular features

| Predictors | Univariate model | | Multivariate model | |
|---|---|---|---|---|
| | Beta (SE) | p-value | Beta (SE) | p-value |
| LTL PRS [‡] | 0.17±0.05 | 0.001 | 0.10±0.04 | 0.009 |
| Telomerase activity | −0.37±0.05 | 9.34E-13 | 0.25±0.06 | 1.32E-05 |
| Proliferation | −0.69±0.04 | 3.30E-46 | −0.80±0.06 | 3.66E-30 |
| Copy number alteration | −0.41±0.05 | 6.36E-16 | −0.23±0.05 | 1.62E-05 |
| Homologous recombination deficiency | −0.4±0.05 | 8.32E-15 | 0.12±0.06 | 0.048 |
| Tumour total mutation burden | −0.28±0.05 | 1.37E-07 | −0.09±0.24 | 0.695 |
| SBS1 | −0.18±0.05 | 0.001 | 0.01±0.05 | 0.827 |
| SBS4 | −0.24±0.05 | 6.36E-06 | 0.04±0.18 | 0.814 |
| SBS5 | −0.27±0.05 | 4.84E-07 | 0.03±0.09 | 0.770 |

SBS (single base substitution DNA mutational signatures); LTL = leukocyte telomere length; PC = principal component; PRS = polygenic risk score.

[*]age of diagnosis represented as beta estimate per 1 unit of SD.

[†]OR per 1 unit of SD.

[‡]LTL PRS is adjusted by first five PC of genetic ancestry in the univariate model.

higher in TCGA-LUAD than in TCGA-LUSC (*Figure 5B*), while higher proliferation rate (*Figure 5C*, $p=1.45 \times 10^{-141}$) and *TERT* activity (*Figure 5D*, $p=1.36 \times 10^{-20}$) were observed in TCGA-LUSC than in TCGA-LUAD cases. Of note, the low RNA levels of the telomere-related genes (less than five read counts), such as *TERT* and *TERC*, in both TCGA-LUAD and TCGA-LUSC tumour samples limited the direct comparison of these genes between these cohorts.

## Discussion

The maintenance of telomere length is one of the hallmarks of cancer, being critical for cell proliferation and genome integrity (*Hanahan and Weinberg, 2011*). Individual differences in telomere length, measured either directly or indirectly by germline determinates, have been linked with multiple diseases, including cancer susceptibility (*Codd et al., 2021*). The measurement of LTL within the UKBB has provided a resource for the development of a more powerful set of genetic instruments that capture a greater proportion of variation in LTL compared to previous studies (*Codd et al., 2021*). We applied genetic determinants of LTL to the largest GWAS of lung cancer to further characterise the role of telomere maintenance in lung cancer aetiology.

Using an MR analysis framework, we confirmed the previously reported relationship between genetically predicted longer LTL and increased risk of lung cancer. Our expanded genetic instrument detected systematic negative pleiotropy, which has not been observed in previous MR studies (*Zhang et al., 2015*; *Haycock et al., 2017*; *Kachuri et al., 2019*). Correcting for this pervasive directional

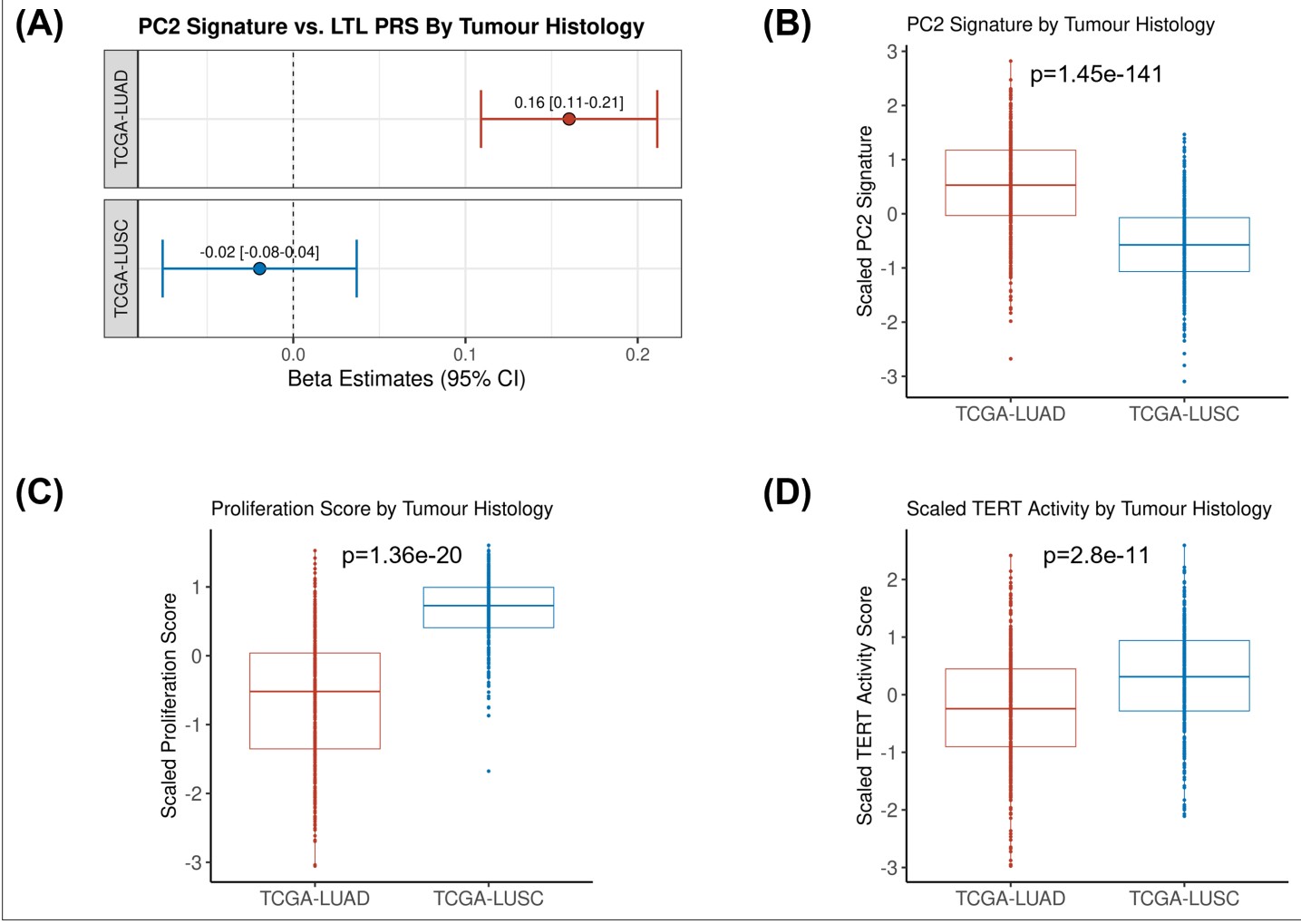

**Figure 5.** Comparing inferred PC2 gene expression signature by lung cancer histological subtypes. (**A**) Leukocyte telomere length (LTL) polygenic risk score (PRS) association with the 10-gene expression signature of PC2 in lung adenocarcinoma (The Cancer Genome Atlas [TCGA]-LUAD, N=343) and squamous cell carcinoma (TCGA-LUSC, N=338) cases from TCGA dataset. Results are expressed as beta estimate per SD increase in genetically predicted LTL. Linear regression model adjusted by sex, age, and PC1-5 (genetic ancestry) covariates, PC2 signature as outcome. Statistically significant associations with -values<0.05. Values per SD of (**B**) PC2, (**C**) proliferation score, and (**D**) telomerase/*TERT* activity gene expression signatures by lung cancer histological subtypes (TCGA-LUAD and TCGA-LUSC). p-Values derived from Student's t-tests.

The online version of this article includes the following figure supplement(s) for figure 5:

**Figure supplement 1.** Generating inferred PC2 signature based on RNA-sequencing data.

bias resulted in substantially larger effects of LTL on risk of lung adenocarcinoma and lung cancer in never smokers, implying that LTL may even be more important to these phenotypes than previously estimated (*Zhang et al., 2015*; *Haycock et al., 2017*; *Kachuri et al., 2019*). Our observations in never smokers were also supported by multivariate MR analyses where adjustment for smoking did not attenuate the effect of LTL on lung cancer susceptibility. We used MVMR analysis (*Sanderson et al., 2019*) to assess the potential that factors, such as BMI, smoking, alcohol use, and other obesity related factors, may account for the association between LTL and lung cancer. While the influence of alternative unknown potential confounding factors cannot be excluded, the association of LTL with lung cancer risk was materially unaltered following adjustment for a range of potential confounders considered in our MVMR analyses.

Colocalisation analyses for the variants selected as the MR genetic instrument highlighted shared genetic signals between LTL and lung adenocarcinoma, including loci near genes related to telomere length maintenance (*TERT*, *TERC*, and *OBFC1*) and three genetic loci not previously linked

to lung cancer susceptibility (*POLI*, *PRPF6*, and *MPHOSPH6*). The lung cancer risk allele of the *MPHOSPH6* sentinel variant (rs2303262) was associated with longer LTL, reduced pulmonary function, and increased *MPHOSPH6* gene expression in lung tissue. *MPHOSPH6* encodes an enzyme associated with the RNA exosome complex where it modulates RNA binding activity. *PRPF6* in 20q13.33 is involved in androgen binding and has been shown to promote colon tumour growth via preferential splicing of genes involved in proliferation (*Adler et al., 2014*). *POLI* is a member of the Y-family of DNA damage-tolerant polymerases involved in translesion synthesis (*McIntyre, 2020*). As part of its role in DNA repair and replication stress, *POLI* interacts with *TP53* to bypass barriers during DNA replication, which may confer a pro-survival effect to stem cells and cancer cells (*Guo et al., 2021*).

Despite the limited evidence of colocalisation between several loci important for telomere maintenance and lung cancer, the MR analyses restricting to the non-colocalised LTL SNPs pointed out that these loci might lie in the same causal pathway of LTL and lung cancer, highlighting the heterogeneity in the genetic effects of LTL loci.

We additionally identified the relationship between genetic determinants of LTL and a specific gene expression component in lung adenocarcinoma tumours. The aspect of this component associated with longer LTL was also associated with demographic and clinical features, such as never smoking, female, and early-stage tumours compared with other lung adenocarcinoma patients, which is an interesting but not completely understood lung cancer strata. This expression component also tended to be related to genomic features related to genomically stable tumours and strikingly associated with cell proliferation score, implying that this component might be a proxy for this feature. These results appear consistent with the canonical role of telomere length in preserving genome stability and cell proliferation (*de Lange, 2009*).

A plausible explanation for why long LTL was associated with an increased risk of lung cancer might be that individuals with longer telomeres have lower rates of telomere attrition compared to individuals with shorter telomeres. Given a very large population of histologically normal cells, even a very small difference in telomere attrition would change the probability that a given cell is able to escape the telomere-mediated cell death pathways (*Aviv et al., 2017*). Such inter-individual differences could suffice to explain the modest lung cancer risk observed in our MR analyses. However, it is not clear why longer TL would be more relevant to lung adenocarcinoma compared to other lung cancer subtypes. A suggestion may come from our observation that longer LTL is related to genomically stable lung tumours (such as lung adenocarcinomas in never smokers and tumours with lower proliferation rates) but not genomically unstable lung tumours (such as heavy smoking related, highly proliferating lung squamous carcinomas). One possible hypothesis is that histologic normal cells exposed to highly genotoxic compounds, such as tobacco smoking, might require an intrinsic activation of telomere length maintenance at early stages of carcinogenesis that would allow them to survive, and therefore, genetic differences in telomere length are less relevant in these cells. By contrast, in more genomically stable lung tumours, where TL attrition rate is more modest, the hypothesis related to differences in TL length may be more relevant and potentially explain the heterogeneity in genetic effects between lung tumours. Alternately, we also note that the cell of origin may also differ, with lung adenocarcinoma postulated to be mostly derived from alveolar type 2 cells, the squamous cell carcinoma is from bronchiolar epithelium cells (*Sainz de Aja et al., 2021*), possibly suggesting that LTL might be more relevant to the former.

One surprising finding from the genetic analysis was that despite the robust and large effects of LTL on lung cancer risk observed in the MR analyses, the genetic correlation between LTL and lung cancer was effectively null. The LDSC approach considers genetic variants across the entire genome, whereas the MR approach preferentially selects variants based on their association with LTL, restricting to those that achieved genome-wide significance. One possibility for the lack of genetic correlation between LTL and lung cancer is that genetic variants may differ in the direction that they influence these traits, and we used the smoking behaviour traits to exemplify that. For example, the subgroup of genetic variants noted at genome-wide significance from LTL studies was associated with increased LTL and lung cancer risk. However, the subgroup related to smoking behaviours which, in turn, are linked with increased lung cancer risk, tends to decrease LTL. If such opposing effects were widespread across the genome, it could account for the lack of genetic correlation between LTL and lung cancer estimated by LDSC and highlights the complex nature of the genetic variants that determine LTL and lung cancer risk.

Some limitations of this study should be acknowledged. A limited sample size might have limited the detection of an association between lung small-cell carcinoma and LTL. Our colocalisation approach is generally more conservative and may fail to accurately determine the posterior probability for shared genetic signals in the presence of multiple independent associations in a given locus (*Hukku et al., 2021*), which may be a reasonable explanation for the lack of colocalisation observed at *RTEL1* locus, and we stress that many of the variants that are COLOC negative are likely to be associated with lung cancer. Furthermore, the relatively small sample size of the lung adenocarcinoma cohort from TCGA may have reduced the power of our study, and larger cohorts of expression profiles tumours will be necessary to validate and explore some of our findings. The potential limitations such as collider bias within the lung adenocarcinoma case only study design should also be considered.

In conclusion, we describe an association between long genetically predicted LTL and lung cancer risk, which provides insights into how telomere length influences the genetic basis of lung cancer aetiology, including never smoking and female lung adenocarcinoma, which is an enigmatic subset of lung cancer. By using a novel framework to explore the biological implications of genetically complex traits, we unravel one gene expression component, highly correlated with proliferation rate score and other genomic stability-related features, associated with LTL in lung adenocarcinoma tumours. As far as we are aware, this is the first time an association between a PRS related to an aetiological factor, such as telomere length, and a particular expression component in the lung tumours is reported. Our findings suggest that lung adenocarcinoma patients with longer LTL might have more genomically stable tumours than the ones with shorter LTL, shedding some light on telomere biology in those tumours.

## Acknowledgements

We would like to acknowledge the TCGA Research Network (https://www.cancer.gov/tcga) and the contribution of specimen donors and research groups involved in this resource. We also would like to acknowledge the ILCCO consortium, the participants of the UK biobank and GTEx project and the supporting bodies (https://commonfund.nih.gov/GTEx), specimen donors, and research groups. This work was supported by the Institut National du Cancer (INCa) (GeniLuc 2017–1-TABAC-03-CIRC-1 - [TABAC 17-022]), NIH/NCI, INTEGRAL NIH 5U19CA203654-03, Cancer Research UK (grant number C18281/A29019), the France Génomique National infrastructure, funded as part of the « Investissements d'Avenir » program managed by the Agence Nationale pour la Recherche (contract ANR-10-INBS-09). Christopher Amos is a Research Scholar of the Cancer Prevention Institute of Texas ( RR170048). The work of Ricardo Cortez Cardoso Penha reported in this paper was undertaken during the tenure of an IARC Postdoctoral Fellowship at the International Agency for Research on Cancer. Linda Kachuri is supported by funding from the National Institutes of Health (K99CA246076). Philip Haycock is supported by funding from Cancer Research UK (C18281/A29019).

## Additional information

### Funding

| Funder | Grant reference number | Author |
| --- | --- | --- |
| Terry Fox Foundation | IARC Postdoctoral Fellowship | Ricardo Cortez Cardoso Penha |
| National Institutes of Health | K99CA246076 | Linda Kachuri |
| Cancer Prevention and Research Institute of Texas | RR170048 | Christopher Amos |
| Cancer Research UK | K99CA246076 | Philip C Haycock |
| Cancer Research UK | C18281/A29019 | Philip C Haycock |
| Institut National Du Cancer | GeniLuc 2017–1-TABAC-03-CIRC-1 - [TABAC 17-022] | Philip C Haycock |

| Funder | Grant reference number | Author |
|---|---|---|
| National Cancer Institute | INTEGRAL NIH 5U19CA203654-03 | Philip C Haycock |
| Agence Nationale pour la Recherche | ANR-10-INBS-09 | Philip C Haycock |

The funders had no role in study design, data collection and interpretation, or the decision to submit the work for publication.

## Author contributions

Ricardo Cortez Cardoso Penha, Conceptualization, Data curation, Formal analysis, Investigation, Visualization, Methodology, Writing – original draft, Writing – review and editing; Karl Smith-Byrne, Data curation, Formal analysis, Investigation, Methodology, Writing – original draft, Writing – review and editing; Joshua R Atkins, Formal analysis, Methodology, Writing – original draft, Writing – review and editing; Philip C Haycock, Siddhartha Kar, Paul Brennan, Rayjean J Hung, Writing – review and editing; Veryan Codd, Nilesh J Samani, Christopher Nelson, Maja Milojevic, Aurélie AG Gabriel, Data curation, Writing – review and editing; Christopher Amos, Funding acquisition, Writing – original draft, Writing – review and editing; Linda Kachuri, Formal analysis, Investigation, Writing – original draft, Writing – review and editing; James D Mckay, Conceptualization, Formal analysis, Supervision, Funding acquisition, Investigation, Methodology, Writing – original draft, Project administration, Writing – review and editing

## Author ORCIDs

Ricardo Cortez Cardoso Penha  http://orcid.org/0000-0002-2847-1993
Philip C Haycock  http://orcid.org/0000-0001-5001-3350
James D Mckay  http://orcid.org/0000-0002-1787-3874

## Ethics

Human subjects: Mendelian Randomization analyses did not require ethical approval as it used secondary, genome-wide association data from studies that obtained informed consent from all participants and ethical approval from review boards and/or ethics committees. Individual level data, accessed under accession number phs000178.v11.p8 and project application #2731, was subject to the ethnical policies that govern the Cancer Genome Atlas. These policies can be found at https://www.cancer.gov/about-nci/organization/ccg/research/structural-genomics/tcga/history/policies.

## Decision letter and Author response

Decision letter https://doi.org/10.7554/eLife.83118.sa1
Author response https://doi.org/10.7554/eLife.83118.sa2

# Additional files

## Supplementary files

• Supplementary file 1. Supplementary tables (a-m). (a) SNPs selected as leukocyte telomere length (LTL) instrument for the Mendelian randomisation (MR) analyses. (b) MR analyses across methods for the 144 LTL SNPs. (c) Test for directional pleiotropy using MR Egger for the 144 LTL SNPs. (d) Heterogeneity tests for the 144 LTL SNPs. (e) Leave-one out analyses for the 144 LTL SNPs. (f) Multivariable MR analyses. (g) Multivariable MR analyses for LTL and lung cancer adjusting for obesity- and alcohol-related traits. (h) Summary of the colocalisation results for the LTL instrument based on 144 variants. (i) Summary of the multi-trait colocalisation results for the 12 colocalised genetic loci. (j) Top 500 genes positively correlated with the first five principal components. (k) Top 500 genes negatively correlated with the first five principal components. (l) Pathway analysis on the top 500 genes positively and negatively correlated with the first five principal components based on RNA-seq data of 343 ADE cases from The Cancer Genome Atlas (TCGA) dataset using Gene Set Enrichment Analysis (GSEA). (m) Association of LTL polygenic risk score (PRS) with the clinical and molecular features of lung adenocarcinoma tumours.

• MDAR checklist

## Data availability

Lung cancer GWAS summary statistics obtained from ILCCO can be accessed by the database of Genotypes and Phenotypes (dbGAP) under accession phs000876.v1.p1. The GWAS summary statistics for tobacco-smoking behaviors (GSCAN: https://conservancy.umn.edu/handle/11299/201564), LTL (https://figshare.com/s/caa99dc0f76d62990195), and GTEx version 8 (downloaded via GTEx google cloud resource) are publicly available. Germline data of TCGA cohorts were accessed by dbGAP under accession number phs000178.v11.p8 and project application #2731. RNA-sequencing data from TCGA cohorts were retrieved from GDC open-access data portal (https://portal.gdc.cancer.gov/) using TCGAbiolinks package in R. TCGA-related data are publicly available as described in the data section. The code for LDSC analysis is available at: https://github.com/bulik/ldsc/wiki/Heritability-andGenetic-Correlation. The codes used in this study for two-sample MR, colocalisation, multi-trait colocalisation, and principal component analyses are available at https://github.com/ricardocortezcardoso/Telomere_Length_Code, (*Penha, 2023a*, copy archived at swh:1:rev:f365df300919c46bb99a96b4040d90576fc878e2). Plots were created using R packages 'meta' (v5.5, forest plots), 'corrplot' (v0.92, correlation matrix), and 'ggplot2' (v3.3.6). The R package to generate stackplots for visualisation of the multi-trait colocalisation results is available at https://github.com/jrs95/gassocplot, (*Penha, 2023b*, copy archived at swh:1:rev:ae6a59dff2e43d39eead3d483af1d50f151c3d5b).

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

## Appendix I

### Additional methods: Datasets

#### UK Biobank

The UK Biobank (UKBB) is a large cohort with more than 500,000 individuals from the United Kingdom who were recruited between 2006 and 2010, as previously described (*Bycroft et al., 2018*). At the time of recruitment, individuals provided electronic signed consent, answered questions regarding socio-demographic, lifestyle, health-related factors, and physical measures. The genetic data were obtained using the UK BiLEVE and the UKBB Axiom arrays. UKBB has ethical approval from the National Information Governance Board for Health and Social Care and the North-West Multicentre Research Ethics Committee (Ref: 11/NW/0382).

The latest genome-wide association study (GWAS) summary statistics of leukocyte telomere length (LTL; https://figshare.com/s/caa99dc0f76d62990195) were performed on 464,716 UKBB participants of European ancestry with 19.4 million imputed variants ( minor allele frequency$\geq$1% and imputation quality score $\geq$0.3), adjusting by age, sex, array, and the first 10 principal components. LTL was measured from UKBB baseline samples, and values were log-transformed and Z-standardised.

The forced expiratory volume in 1 s ($FEV_1$) and forced vital capacity (FVC) GWAS summary statistics were conducted on 372,750 and 370,638 individuals of European genetic ancestry, respectively, who had spirometry data and sample passed quality control of genetic data (Hardy–Weinberg equilibrium at $p>1 \times 10^{-5}$, MAF >0.5%, and imputation quality score $\geq$0.3). Linear regression models were applied to standardised Z-scores for $FEV_1$ and FVC, adjusting by age, sex, genotyping array, and 15 principal components. These GWAS summary statistics are available by the corresponding authors of the original article upon reasonable request (see *Kachuri et al., 2020*).

#### International lung cancer consortium

The International Lung Cancer Consortium (ILCCO) is an international group of lung cancer researchers for sharing comparable data from ongoing lung cancer case-control and cohort studies. The lung cancer GWAS summary statistics used in the current work is derived from a collaborative effort between Transdisciplinary Research of Cancer in Lung of the ILCCO (TRICL-ILCCO) and the Lung Cancer Cohort Consortium (LC3). All participants in these studies signed an informed consent, approved by the local internal review board or ethics committee.

The lung cancer GWAS summary statistic (29,266 cases and 56,450 controls of European descent; *McKay et al., 2017*) is meta-analysis of the combined imputed genotypes from OncoArray series (14,803 cases and 12,262 controls) and previous lung cancer GWAS (IARC, MDACC, SLRI, ICR, Harvard, NCI, Germany, and deCODE studies; 44,188 controls and 14,436 cases). The meta-analysis only considered imputed variants with MAF >1%, imputation quality score $\geq$0.3, and with little evidence for heterogeneity in effect between the studies (Cochran's Q statistic: p>0.05), adjusting by sex, age, and principal components. Additionally, analyses were conducted by lung cancer histological subtype (adenocarcinoma [11,273 cases and 55,483 controls], squamous cell carcinoma [7426 cases and 55,627 controls], and small-cell carcinoma [2664 cases and 21,444 controls]). Of note, histological subtype information was not available for all studies.

#### The GWAS and Sequencing Consortium of Alcohol and Nicotine use consortium

The GWAS and Sequencing Consortium of Alcohol and Nicotine use dataset is derived from an international genetic association meta-analysis consortium for tobacco smoking and alcohol consumption traits across 30 studies and up to 1.2 million individuals (*Liu et al., 2019*). For the GWAS summary statistics used in the current study, UKBB participants were excluded from the meta-analysis, and only genetic variants present in more than 19 studies, MAF >1%, and imputation quality score >0.3 were considered. The smoking traits were defined as follow: cigarettes per day (CPD; N=337,334; continuous variable defined as average number of cigarettes smoked per day, either as a current smoker or former smoker), smoking initiation (SmkInit; N=1,232,091; categorical variable defined as ever smokers versus never smokers), smoking cessation (SmkCes; N=547,219; categorical variable defined as current smokers versus former smokers), and age of initiation (AgeSmk; N=341,427; continuous variable defined as the age at which an individual first became a regular smoker). Age, sex, and genetic principal components were used as covariates in all analyses.

GWAS summary statistics were downloaded from https://conservancy.umn.edu/handle/11299/201564.

The Genotype-Tissue Expression (GTEx) project data is a project that aims to investigate the effects of genetic variants in transcriptome and their impact on regulatory mechanisms of a traits and diseases (*GTEx Consortium, 2020*). The GTEx data contain 15,201 RNA-sequencing samples from 49 tissues of 838 postmortem donors. For the current study, we accessed version 8 of the GTEx program to obtain the effect estimates and risk alleles of eQTL genetic variants (in normal lung tissue) around the queried SNPs for the multi-trait colocalisation analyses.

## The Cancer Genome Atlas

The Cancer Genome Atlas (TCGA) is currently the biggest resource of cancer genomics, with more than 20,000 primary cancer and matched normal samples molecularly characterised across 33 cancer types with available demographic and clinical data.

For the current study, the access for the germline data of TCGA dataset was obtained via TCGA project #2731 via dbGAP (phs000178.v11.p8). Genotyping and imputation from TCGA data were performed in-house and described elsewhere (*Gabriel et al., 2022*). Briefly, germline variants with low genotyping call rate (<97%), MAF <1%, strong deviation (p-value<10$^{-8}$) from the Hardy–Weinberg equilibrium, and allele frequency differing more than 20% from 1000 Genomes (1000 G) within European ancestry group were removed. Phasing was performed using Eagle (v2.4.1; *Gabriel and Lipinski, 2021*) with the 1000 genomes phase 3 reference panel, and Minimac4 (v1.0.1; *Gabriel and Lipinski, 2021*) was used to perform the imputation (window size of 500 kb; *Das et al., 2016*). Samples with discrepancies between genetic inferred and self-reported sex as well as with high inter-sample relatedness (PI_HAT >0.185%) were removed.

Of the 480 lung adenocarcinoma cases with genotype data, 364 cases had available tumour RNA-sequencing data (more details in Data availability section). We excluded 20 cases due to missing information for smoking status (ever smokers versus never smokers) from the downstream analyses. After summarising the normalised log-transformed read counts for all genes in two dimensions, one sample was removed due to extreme gene expression profile in comparison with all the lung adenocarcinoma tumours. Thus, 343 lung adenocarcinoma tumour samples were selected for the principal component analysis and association with LTL polygenic risk score (PRS), molecular, clinic, and demographic variables of TCGA dataset.

