## [Editor Report]

This study is of interest to epidemiologists and geneticists studying the association between telomere length and lung cancer risk. This work provides useful insight into risk factors for lung cancer. Overall, the results of this study are solid, as the genetic instrument used here is better powered and the battery of Mendelian randomization analysis makes this broad set of results convincing compared to previous work.

---

## [Decision Letter]

**Decision letter after peer review:**

Thank you for submitting your article "Common genetic variations in telomere length genes and lung cancer: a Mendelian Randomization study and its novel application in lung tumor transcriptome" for consideration by *eLife*. Your article has been reviewed by 2 peer reviewers, and the evaluation has been overseen by a Reviewing Editor and a Senior Editor. The following individual involved in the review of your submission has agreed to reveal their identity: Ben Voight (Reviewer #2).

As is customary in *eLife*, the reviewers have discussed their critiques with one another. What follows below is the Reviewing Editor's edited compilation of the essential and ancillary points provided by reviewers in their critiques and in their interaction post-review. Please submit a revised version that addresses these concerns directly. Although we expect that you will address these comments in your response letter, we also need to see the corresponding revision clearly marked in the text of the manuscript. Some of the reviewers' comments may seem to be simple queries or challenges that do not prompt revisions to the text. Please keep in mind, however, that readers may have the same perspective as the reviewers. Therefore, it is essential that you attempt to amend or expand the text to clarify the narrative accordingly.

Essential revision:

The reviewers provided a set of critiques that are manageable in scale. Please respond to all of the reviewer comments.

*Reviewer #1 (Recommendations for the authors):*

Figures are difficult to read as they are poor resolution.

*Reviewer #2 (Recommendations for the authors):*

1. Resolution is not great on some of the figures – made reading them a little bit hard. I trust the authors will provide higher-resolution versions in the proofing phases!

2. Figure 1B – Not sure what this overall means/how to be interpreted – I think this is too much data plotted on the same figure for my taste.

3. I think the authors should create a supplementary table that enumerates the list of SNPs used for their instrument, along with annotated info about the SNP (pos, alleles, allele frequency), effect estimate, p-value report; then, for each SNP, the relevant outcome data of interest in the paper (effect sizes, se, P-value). That way, it is perfectly clear what SNPs are used and what estimated effects are used as input for MR experiments.

4. I'm a little confused about the colocalization analysis:

a. Suppl. Table 6: it looks to me like there are annotation mislabelling issues.

b. Also it isn't clear in this table which LTL loci have /any/ evidence of lung cancer association, i.e., PP1/2 vs. PP3.

For these, suggest checking the table and adding a column for the top lung cancer association (and SNP) in the interval tested by coloc.

c. But also confusing somewhat that there appear to be multiple TERT signals and those with colocalization but are basically on top of one another whilst still being LD independent (e.g., rs61748181 and rs33977403).

Overall here it looks like there may be some differences between genetic variants selected at this locus compared between Codd et al. and here. It could be these choices make sense and are reasonable, but might be worth looking back at differences to see if different choices drive the results they see.

5. In methods describing the Lung cancer data, are histologic data a complete subset of the overall data? It looks like there are shared controls (? a small limitation if so), but why is the number of controls in two strata greater than the overall? The cases totals are also less than overall, are 'lost' cases here simply because a subtype isn't specified? Perhaps the authors can clarify this a bit in methods.

6. Given the sample size differences, what effect sizes were the authors powered to observe in the Lung Squamous cell carcinoma and Lung Small Cell Carcinoma groups? Should this be interpreted as a false negative due to lack of power OR were the authors reasonably powered to observe (say) the estimated effect size overall in the univariate or epidemiological correlation data (but failed to see that effect), consistent with apparent heterogeneity in these strata?

7. As the author indicates, the MR Egger results (Table S2) suggest substantial 'negative' confounding. This is a little confusing – in my experience, I have seen negative confounding for sure previously, but this is probably some of the stronger effects that I've seen before. I don't have any great advice about how to look at this in detail, other than perhaps through MVMR, or by looking carefully at the IV regression plot to see if there are outliers that are somehow strongly influencing MR-Egger.

8. Inclusion of an epidemiological correlational data baseline between LTL >> lung cancer to compare to the MR estimates would also add value.

[Editors' note: further revisions were suggested prior to acceptance, as described below.]

Thank you for resubmitting your work entitled "Common genetic variations in telomere length genes and lung cancer: a Mendelian Randomization study and its novel application in lung tumour transcriptome" for further consideration by *eLife*. Your revised article has been evaluated by a Senior Editor and a Reviewing Editor.

The manuscript has been improved but there are some remaining issues that need to be addressed, as outlined below:

New revisions required:

*Reviewer #2 (Recommendations for the authors):*

The authors have responded to most comments reasonably.

However, I still have some concerns about the given report. I know the collection of these points below are a little extra work -- I am honestly not trying to "add on" but do think these experiments are important to bring out the robustness of the author's claim, add value to the extant literature, and interpretation at these key points around the MR they describe.

1. First, focused around my original comment #2 (in the author's response, their point #6) on multivarible MR experiments. To my point that there are actually multiple potential confounders – including smoking, BMI, and waist-hip ratio that could potentially confound the interpretation that LTL explains the lung-cancer associations, the authors performed one experiment focused on BMI. They note that for each cancer trait there appears to be no impact on their primary association between LTL and Lung cancer data when considering BMI.

However, what they present is still a bit inadequate from my point of view, in the following ways.

1a. The authors perform a form of this MVMR which is a little 'less' good in my view for this -- they build a new instrument based on SNPs ascertained for association with BMI, and then include that in the MVMR regression analysis. This is not how I would do this -- this result describes a "joint" model for the effect of exposures (BMI, LTL) on outcome (Lung cancer). This version does not regress out any potential association of the SNPs *selected for association with LTL* has with the potential confounder. In this case, if the LTL variants had strong association with the confounder (e.g., BMI), I don't think that this would be fully accounted for.

Instead, the authors should take the n=144 SNPs selected for the LTL instrument, identify associations from extant data with the putative confounder of interest (e.g., BMI), create an instrument for the genetically mediated effects of that confounder, and include that as an instrument for covariate confounding.

I.e., "regress out" any effect that the LTL instrument might have on the confounder, and report adjusted effects of the LTL instrument on cancer risk.

Yes: this may NOT be a very strong instrument for the confounder -- but that's not the point here. The point is to determine the effect of genetically mediated LTL on cancer adjusting for any effect that instrument the authors built for it has on the confounder.

1b. Furthermore, considering a single confounder here is really suboptimal. As indicated in the previous comments and as the authors know, smoking behavior and other anthropomorphic measures (e.g., waist-hip ratio) are also correlated with everything here. So the authors should really consider an expanded list of confounders for this to be clearly convincing that LTL really does operate as a causal exposure independently of these other risk factors.

1c. In addition to experiments that are pairwise, they should also consider performing an experiment with all potential confounders included in the model

Cancer ~ LTL + BMI + WHR + SMOKING + …..

I.e., can the authors convincingly demonstrate that the observed relationship between LTL and these lung cancer outcomes is not explained by a battery of obvious confounders individually or as a collective

1d. As suggested earlier – I think value and novelty would be added if the authors really took this on and explored this space a bit more -- look hard and think carefully about what confounders really *COULD* explain this result and work hard to refute that those could explain these observations

I do think a robust set of experiments should be included in the primary report beyond smoking, either way (in the supplement). I think the authors report the BMI result in the rebuttal response to me but don't add those back to the paper. I'd honestly like to see a healthy version of that added to the paper, but that's my taste on this point.

2. On my previous point #4 (in the rebuttal, point #8), I asked the authors to consider a subset MR analysis which focused on those where the genes/signals map quite obviously on known telomere length biology and represent some of the strongest signals. The authors respond that they can't actually do this analysis, suggesting that using the data for co-localization somehow invalidates an MR sub-set analysis.

I thought about this a little bit and while I admit that I can be daft sometimes I honestly do not understand what basis – empirical or theoretical – this argument has to somehow drive positive or negative biases here, i.e., why they need an "out of sample" set to perform the MR after they used this data to perform colocalization. If there's specific literature or reasoning the authors would like to articulate this in more detail, that would certainly be appreciated!

My intuition is that this actually isn't much of an issue -- performing subset analyses for different reasons (biological or otherwise) is actually pretty reasonable and there are MR methods that actually try to cluster / subset instruments used.

This all goes to interpretation and as a sensitivity analysis -- if the result is driven entirely by variation at TERT, TERC, OBFC1 but not obviously by other variation that influences telomere length, then I personally think that is worth trying to understand and report. Analytically, this is a *trivial* experiment to perform in the UVMR space.

3. Regarding the rebuttal point #13, which involved my point about power for discovery in the subset given the sample size, the authors state that there are sample size difference but then write something about confidence intervals overlapping that I don't quite understand.

What I think the authors SHOULD do with respect to the tumor histology analysis is to report a credible set based on power calculation which determine what range of causal effect sizes the Lung Squamous CC and Lung Small CC were powered to discover (at an α type-1 error rate) of 5% and/or 1%, say, given the instruments used and sample sizes involved here.

This is actually a very easy thing to do analytically with some assumptions – back of envelope -- you could check out what formula I pulled together in PMID: 25165093 to get the distribution under the alternative; I believe there are also MR power calculators out there as web tools. You easily do this with some R calculations in a straight forward way.

I think the authors should be able to articulate is that effect sizes (say) of OR=1.6 or better would be discoverable in those lung cell subsets that basically catch the null hypothesis. Or, that those sets were well powered to discover ORs that were even smaller than that. This would then give you a quantitative assessment that this isn't simply a false-negative result, but that there is putative heterogeneity here. But if they are small enough that they AREN'T particularly well powered, then that's a pretty important interpretive point on this plot.

---

## [Author Response]

Essential revision:The reviewers provided a set of critiques that are manageable in scale. Please respond to all of the reviewer comments.Reviewer #1 (Recommendations for the authors):Figures are difficult to read as they are poor resolution.

We apologize for these lower resolution figures. All figures have been updated in higher resolution (600 dpi) in the revised manuscript.

Reviewer #2 (Recommendations for the authors):1. Resolution is not great on some of the figures – made reading them a little bit hard. I trust the authors will provide higher-resolution versions in the proofing phases!

We apologize for these lower resolution figures. All figures have been updated in higher resolution (600 dpi) in the revised manuscript.

2. Figure 1B – Not sure what this overall means/how to be interpreted – I think this is too much data plotted on the same figure for my taste.

The figure 1B sought to provide insights on the possible reasons why there were not genome-wide genetic correlations observed between lung adenocarcinoma (or any lung cancer histological subtype) and genetically predicted leukocyte telomere length.

We are highlighting in this figure that only a small subset of LTL variants is associated with longer telomere length and higher risk of lung adenocarcinoma development on one hand, but variants also related to lung cancer risk such as smoking behavior traits are associated in the opposite direction. We speculated that a possibility for the lack of genetic correlation between LTL and lung cancer may be these opposing effects that may explain the lack of global genetic correlation between TL and lung adenocarcinoma.

We have re-worded the figure legend to make this figure clearer to the reader and further highlighted this aspect and link to this figure in the discussion.

3. I think the authors should create a supplementary table that enumerates the list of SNPs used for their instrument, along with annotated info about the SNP (pos, alleles, allele frequency), effect estimate, p-value report; then, for each SNP, the relevant outcome data of interest in the paper (effect sizes, se, P-value). That way, it is perfectly clear what SNPs are used and what estimated effects are used as input for MR experiments.

Thanks, this reviewer for the suggestion. We added supplementary file 1a listing all the SNPs used as LTL instruments in the current study in the revised manuscript.

4. I'm a little confused about the colocalization analysis:a. Suppl. Table 6: it looks to me like there are annotation mislabelling issues.

We apologize for the annotation mislabelling issues, and the table was corrected in the revised manuscript. The genetic variants were annotated using ANNOVAR tool and the dbSNP150 database in the supplementary file 1g of the revised manuscript. Of the 144 SNPs from the LTL instrument, 11 were replaced by proxies (R^2^>0.8, whenever possible, in European population 1000 genomes) in the current study because they were missing in the lung adenocarcinoma GWAS summary statistic. A column was added to specify the proxies in the supplementary file 1g of the revised manuscript.

b. Also it isn't clear in this table which LTL loci have /any/ evidence of lung cancer association, i.e., PP1/2 vs. PP3.For these, suggest checking the table and adding a column for the top lung cancer association (and SNP) in the interval tested by coloc.

We have included the columns with PP1/PP2/PP3 in the supplementary file 1g of the revised manuscript. Only one locus (*RETL1*) displayed high PP3 (0.99). We have added the information of this PP3 levels in our results describing the lack of colocalisation for the variants within *RTEL1* locus (PP4=0) in the result section of the revised manuscript (page 12, lines 260-264).

c. But also confusing somewhat that there appear to be multiple TERT signals and those with colocalization but are basically on top of one another whilst still being LD independent (e.g., rs61748181 and rs33977403).

Again, we apologise for the error in the genetic variant annotation in the former supp table S7/ current supplementary file 1g noted above. This has complicated again the interpretation relative to this point. Supplementary file 1g has been corrected in the revised version of the manuscript. Indeed, there seems to be multiple independent hits at *TERT* locus when using our filtering (r2<0.01).

5. In methods describing the Lung cancer data, are histologic data a complete subset of the overall data? It looks like there are shared controls (? a small limitation if so), but why is the number of controls in two strata greater than the overall? The cases totals are also less than overall, are 'lost' cases here simply because a subtype isn't specified? Perhaps the authors can clarify this a bit in methods.

Indeed, histological subtype information was not available for all studies. We clarify this point in the revised manuscript (page 34, line 755).

6. Given the sample size differences, what effect sizes were the authors powered to observe in the Lung Squamous cell carcinoma and Lung Small Cell Carcinoma groups? Should this be interpreted as a false negative due to lack of power OR were the authors reasonably powered to observe (say) the estimated effect size overall in the univariate or epidemiological correlation data (but failed to see that effect), consistent with apparent heterogeneity in these strata?

Yes, there are differences in sample size and therefore the statistical power within lung cancer strata, nevertheless, the confidence intervals of squamous cell carcinoma and small cell carcinoma did not overlap, implying a distinct association rather than a lack of power.

7. Inclusion of an epidemiological correlational data baseline between LTL >> lung cancer to compare to the MR estimates would also add value.

Unfortunately, we do not have measured telomere length at baseline for most of the case series and we used the LTL MR instrument to infer it.

[Editors' note: further revisions were suggested prior to acceptance, as described below.]

Reviewer #2 (Recommendations for the authors):The authors have responded to most comments reasonably.However, I still have some concerns about the given report. I know the collection of these points below are a little extra work -- I am honestly not trying to "add on" but do think these experiments are important to bring out the robustness of the author's claim, add value to the extant literature, and interpretation at these key points around the MR they describe.1. First, focused around my original comment #2 (in the author's response, their point #6) on multivarible MR experiments. To my point that there are actually multiple potential confounders – including smoking, BMI, and waist-hip ratio that could potentially confound the interpretation that LTL explains the lung-cancer associations, the authors performed one experiment focused on BMI. They note that for each cancer trait there appears to be no impact on their primary association between LTL and Lung cancer data when considering BMI.However, what they present is still a bit inadequate from my point of view, in the following ways.1a. The authors perform a form of this MVMR which is a little 'less' good in my view for this -- they build a new instrument based on SNPs ascertained for association with BMI, and then include that in the MVMR regression analysis. This is not how I would do this -- this result describes a "joint" model for the effect of exposures (BMI, LTL) on outcome (Lung cancer). This version does not regress out any potential association of the SNPs *selected for association with LTL* has with the potential confounder. In this case, if the LTL variants had strong association with the confounder (e.g., BMI), I don't think that this would be fully accounted for.Instead, the authors should take the n=144 SNPs selected for the LTL instrument, identify associations from extant data with the putative confounder of interest (e.g., BMI), create an instrument for the genetically mediated effects of that confounder, and include that as an instrument for covariate confounding. I.e., "regress out" any effect that the LTL instrument might have on the confounder, and report adjusted effects of the LTL instrument on cancer risk. Yes: this may NOT be a very strong instrument for the confounder -- but that's not the point here. The point is to determine the effect of genetically mediated LTL on cancer adjusting for any effect that instrument the authors built for it has on the confounder.1b. Furthermore, considering a single confounder here is really suboptimal. As indicated in the previous comments and as the authors know, smoking behavior and other anthropomorphic measures (e.g., waist-hip ratio) are also correlated with everything here. So the authors should really consider an expanded list of confounders for this to be clearly convincing that LTL really does operate as a causal exposure independently of these other risk factors.1c. In addition to experiments that are pairwise, they should also consider performing an experiment with all potential confounders included in the model Cancer ~ LTL + BMI + WHR + SMOKING + … i.e., can the authors convincingly demonstrate that the observed relationship between LTL and these lung cancer outcomes is not explained by a battery of obvious confounders individually or as a collective.1d. As suggested earlier – I think value and novelty would be added if the authors really took this on and explored this space a bit more -- look hard and think carefully about what confounders really *COULD* explain this result and work hard to refute that those could explain these observations.I do think a robust set of experiments should be included in the primary report beyond smoking, either way (in the supplement). I think the authors report the BMI result in the rebuttal response to me but don't add those back to the paper. I'd honestly like to see a healthy version of that added to the paper, but that's my taste on this point.

We thank the reviewer for their thorough and detailed review of our methods. We would like to take this opportunity to clarify that the standard MVMR analysis does consider multiple instrumental variables. The analyses that we have performed indeed account for the associations of LTL instruments with other variables. These established methods are reviewed in detail here (PMID: 30535378) and we have added this citation to the methods for clarity.

For the BMI example, the inputs for the MVMR analyses were as follows:

Therefore, the output of the MVMR analysis is a reasonable test of the effect of LTL on lung cancer that conditions on the genetic associations between LTL instruments and BMI.

We focused our initial response using BMI as an MVMR analysis for an adiposity-related trait was specifically requested by the reviewer and we felt BMI was reasonable in that context.

We have now expanded this to other traits. These include MVMR analyses for LTL on lung cancer adjusting for waist-to-hip-ratio (WHR), HDL, total triglycerides, blood pressure, smoking, and alcohol intake. These expanded analyses did not result in any material change to our conclusions. We similarly fitted a multivariate model including multiple traits (LC~ LTL+BMI+CPD+drinksperWeek), which similarly noted no material difference to our conclusion that LTL is associated with lung cancer risk. We acknowledge the reviewers point that these results should be included in the manuscript. They were included as supplementary file 1g in the revised manuscript.

We similarly acknowledge the reviewers point that we cannot exclude a potential effect of an unknown confounder. As such, we included the following sentence in the Discussion section of the revised manuscript: “We used MVMR analysis (citation pubmed PMID: 30535378) to assess the potential that factors, such as BMI, smoking, alcohol use and other obesity related factors, may account for the association between LTL and lung cancer. While the influence of alternative unknown potential confounding factors cannot be excluded, the association of LTL with lung cancer risk was materially unaltered following adjustment for a range of potential confounders considered in our MVMR analyses.”

2. On my previous point #4 (in the rebuttal, point #8), I asked the authors to consider a subset MR analysis which focused on those where the genes/signals map quite obviously on known telomere length biology and represent some of the strongest signals. The authors respond that they can't actually do this analysis, suggesting that using the data for co-localization somehow invalidates an MR sub-set analysis.I thought about this a little bit and while I admit that I can be daft sometimes I honestly do not understand what basis – empirical or theoretical – this argument has to somehow drive positive or negative biases here, i.e., why they need an "out of sample" set to perform the MR after they used this data to perform colocalization. If there's specific literature or reasoning the authors would like to articulate this in more detail, that would certainly be appreciated!My intuition is that this actually isn't much of an issue -- performing subset analyses for different reasons (biological or otherwise) is actually pretty reasonable and there are MR methods that actually try to cluster / subset instruments used.This all goes to interpretation and as a sensitivity analysis -- if the result is driven entirely by variation at TERT, TERC, OBFC1 but not obviously by other variation that influences telomere length, then I personally think that is worth trying to understand and report. Analytically, this is a *trivial* experiment to perform in the UVMR space.

We apologise if our initial response was not sufficiently clear. While MR approaches that cluster variants within genetic instruments can be used to identify biological relevant subgroups of genetic predictors (DOI:10.1093/bioinformatics/btaa778), the results generated by this method can be challenging to interpret.

In our study, the genetic variants included in the MR analyses that was used to estimate the association of LTL with lung cancer risk have been associated with LTL (at least to genome wide significance) in an independent large GWAS (Codd et al. NG Nat Genet 2021). We subsequently used COLOC to highlight loci that have a particularly strong evidence of a shared causal association between LTL and lung cancer risk to enrich our understanding of the association of LTL with lung cancer. Our study has not sought to identify a subset of loci that predict LTL that are not associated with lung cancer risk.

We feel that MR analysis based on the stratified MR instrument (COLOC+ *vs.* COLOC-) would greatly complicates the interpretation of the results. Our concern is that to successfully estimate colocalisation, the variants of interest should be associated with both traits (i.e. LTL and lung cancer), particularly when conservative priors are used. By creating an MR instrument based on variants that are COLOC+, we are therefore selecting variants that are strongly associated with lung cancer in this dataset. As such, if that MR instrument was then re-applied into the same dataset, we expect that the instrument will return an estimate biased upwards due to how we selected the variants. We have created a MR instrument based on COLOC+ variants (n=12, variance in LTL explained of 1.05%) and applied that COLOC+ instrument in an MR analysis within our lung cancer dataset. This analysis is presented in the table below. The MR estimates using the COLOC+ instrument had a larger association with lung cancer risk (OR = 3.48 for lung adenocarcinoma (table below)) compared to MR instrument using all variants associated with LTL (OR = 2.43 for lung adenocarcinoma (figure 2)). However, in line with our concerns above, it is unclear to us if this increase in magnitude of association is driven by anything other than the expected greater strength of association for contributing variants and lung cancer risk provoked by the in-sample selection of these variants. We feel that an independent sample would be required to distinguish potential bias from a true biological difference from the effects of the COLOC+ MR instrument, which did not fall within the remit of our study.

**Author response table 1. sa2table1:** 

Exposure	Outcome	Method	P	OR	LCI	UCI
Telemore Length (Colocalised SNPs)	Lung Cancer	Inverse variance weighted	3.93E-06	2.21	1.58	3.10
Telemore Length (Colocalised SNPs)	Lung Adenocarcinoma	Inverse variance weighted	2.91E-07	3.48	2.16	5.60
Telemore Length (Colocalised SNPs)	Lung Squamous Cell Carcinoma	Inverse variance weighted	0.427	1.25	0.72	2.19
Telemore Length (Colocalised SNPs)	Lung Small Cell Carcinoma	Inverse variance weighted	0.691	0.822	0.31	2.19
Telemore Length (Colocalised SNPs)	Lung Ever Smokers	Inverse variance weighted	0.008	1.79	1.16	2.74
Telemore Length (Colocalised SNPs)	Lung Never Smokers	Inverse variance weighted	0.122	3.01	0.74	12.20
Telemore Length (Non-Colocalised SNPs)	Lung Cancer	Inverse variance weighted	2.36E-08	1.51	1.31	1.74
Telemore Length (Non-Colocalised SNPs)	Lung Adenocarcinoma	Inverse variance weighted	5.85E-15	2.38	1.91	2.96
Telemore Length (Non-Colocalised SNPs)	Lung Squamous Cell Carcinoma	Inverse variance weighted	0.293	0.89	0.72	1.10
Telemore Length (Non-Colocalised SNPs)	Lung Small Cell Carcinoma	Inverse variance weighted	0.852	1.03	0.76	1.39
Telemore Length (Non-Colocalised SNPs)	Lung Ever Smokers	Inverse variance weighted	8.35E-06	1.44	1.23	1.69
Telemore Length (Non-Colocalised SNPs)	Lung Never Smokers	Inverse variance weighted	1.03E-03	1.85	1.28	2.66

As suggested by the reviewer, an alternate approach could be the stratification of variants based on biological rational, limiting to genes encoding proteins described in telomere maintenance, for example. However, in this instance, we have performed the COLOC analysis and based on those COLOC results, we have already noted that telomere maintenance genes are those that tend to colocalise (Pg 15 lines 328-330). Therefore, a MR instrument based on the variants located at or near genes involved in telomere maintenance may also be potentially biased because of the observations we have already made.

We also have concerns about variants in the COLOC(-) strata. Here, we might expect a result to be estimates biased downwards (i.e. we could be selecting within this data, a set of variants that are less strongly associated with lung cancer). However, this isn’t that straight forward as COLOC is a generally conservative method and, as stated previously, we employed relatively strict priors. Additionally, COLOC can be a less efficient estimator of shared causal loci where multiple causal variants exist in a region but where only a subset are shared between two or more traits. As such, many loci associated with LTL but without strong evidence for colocalisation in our study may be associated with lung cancer. Moreover, a large proportion of variation in LTL is explained by COLOC(-) strata variants (n=132, variance explained of 2,45%). MR analyses using the COLOC(-) strata variants described in table below do estimate a strong association between LTL ad lung cancer. For these reasons, we feel that interpreting results based on genetic instruments stratified by COLOC status is very challenging for a reader.

We appreciate this reviewer’s point of view and thank them for their input. However, for the complexities related to how we have employed COLOC subsequent to our MR analysis outlined above, we prefer to be conservative and limit our analysis to MR instruments selected from the large LTL from a large GWAS of telomere length.

We highlight the limitations of the COLOC approach (discussion lines 328-329) and have adapted this text further for clarity.

“Our colocalisation approach is generally more conservative and may fail to accurately determine the posterior probability for shared genetic signals in the presence of multiple independent associations in a given locus (45) and we stress that many of the variants that are COLOC negative are likely to associated with lung cancer. Indeed, which may be a reasonable explanation for the lack of colocalisation observed at *RTEL1* locus”.

3. Regarding the rebuttal point #13, which involved my point about power for discovery in the subset given the sample size, the authors state that there are sample size difference but then write something about confidence intervals overlapping that I don't quite understand.What I think the authors SHOULD do with respect to the tumor histology analysis is to report a credible set based on power calculation which determine what range of causal effect sizes the Lung Squamous CC and Lung Small CC were powered to discover (at an α type-1 error rate) of 5% and/or 1%, say, given the instruments used and sample sizes involved here.This is actually a very easy thing to do analytically with some assumptions – back of envelope -- you could check out what formula I pulled together in PMID: 25165093 to get the distribution under the alternative; I believe there are also MR power calculators out there as web tools. You easily do this with some R calculations in a straight forward way.I think the authors should be able to articulate is that effect sizes (say) of OR=1.6 or better would be discoverable in those lung cell subsets that basically catch the null hypothesis. Or, that those sets were well powered to discover ORs that were even smaller than that. This would then give you a quantitative assessment that this isn't simply a false-negative result, but that there is putative heterogeneity here. But if they are small enough that they AREN'T particularly well powered, then that's a pretty important interpretive point on this plot.

Thank the reviewer for this fair suggestion. We performed power calculations to determine the range of LTL causal effects on lung cancer that we are adequately powered to detect, and report these results in Supplementary Figure 2—figure supplement 1 (panel B) and on pages 11 and 18 of the revised manuscript. As seen in Figure 2—figure supplement 1b, our study has >80% power to detect rather modest effects (OR~1.20) for lung cancer overall, adenocarcinoma, and squamous cell carcinoma. For less common subtypes, such as small cell carcinoma and lung cancer in never smokers, we 80% power to detect OR~1.40 and OR~1.50, respectively.